# The impact of malaria-induced neutrophil subset shift and a link to Burkitt lymphoma

Sharon Akinyi Okoth[1,2], Ronald K. Tonui [3,4], Titus K. Maina[5], Eddy Agwati[1,2], Cliff I. Oduor[5], Zachary Racenet[6], Viriato M'Bana[6], Festus M. Njuguna[7], Kibet K. Keitany[7], Daniel Chepsiror[7], Cyrus Ayieko[1], Ann M. Moormann[6], Ann W. Kinyua[2], Catherine S. Forconi [6]*

1 Department of Zoology, Maseno University, Maseno, Kenya, 2 Center for Global Health Research, Kenya Medical Research Institute, Kisumu, Kenya, 3 Department of Pathology, Moi University School of Medicine, Kenya, 4 Academic Model Providing Access to Healthcare (AMPATH), Eldoret, Kenya, 5 Department of Pathology and Laboratory Medicine, Brown University, Providence, Rhode Island, United States of America, 6 Department of Medicine, University of Massachusetts Chan Medical School, Worcester, Massachusetts, United States of America, 7 Moi Teaching and Referral Hospital, Eldoret, Kenya

* Catherine.Forconi@umassmed.edu

## Abstract

Burkitt lymphoma (BL) is an aggressive B-cell lymphoma that remains a leading cause of childhood cancer mortality in sub-Saharan Africa. Although the epidemiological link between *Plasmodium falciparum (Pf)* malaria and BL has been established, our understanding of the underlying immunological mechanisms conducive to tumorigenesis is incomplete. To address a noted gap in our knowledge of the immune landscape, we conducted a prospective study to profile neutrophil subsets from children with different exposure histories to *Pf*-malaria and children diagnosed with BL from Western Kenya, along with healthy malaria low-exposed Kenyan adults. Using multiparameter flow cytometry, we characterized neutrophils by expression of CD15, CD16, CD10, CD11b, CD182, CD184, and CD62L and found that malaria-exposed children exhibited increased frequencies of aged neutrophil subsets, accompanied by a reduction in the mature active subset frequencies compared to malaria low-exposed children. Notably, a positive correlation (rs = 0.7; p < 0.0001) was observed in immature neutrophils between malaria-exposed healthy and BL children, revealing a possible similar expansion of this subset in both groups. These findings suggest a malaria-associated expansion of the immature neutrophil subset. While functional assays were not performed in this study, previous reports indicate that immature neutrophils can exhibit tumor-promoting functions. Therefore, the observed shift in neutrophil profiles may reflect phenotypic changes associated with malaria exposure that could contribute to a permissive environment for BL.

**Data availability statement:** All relevant data are within the manuscript and its Supporting Information files.

**Funding:** NIH R01 CA189806 (Moormann). The funders had no role in study design, data collection and analysis, decision to publish, or preparation of the manuscript.

**Competing interests:** The authors have declared that no competing interests exist.

## Introduction

Endemic Burkitt lymphoma (BL) is an aggressive childhood B-cell lymphoma that accounts for 50%–70% of childhood cancers in sub-Saharan Africa [1,2], where malaria is also endemic [3]. Chronic malaria exposure predisposes children to BL by inducing impairments in host immune responses [4] such as reduced IFN-γ T cell responses, suppression of T cell responses, as well as a skewed T cell immune profile [5–7]. The result of these alterations is the flawed immune surveillance against Epstein-Barr Virus (EBV), a well-established oncogenic cofactor in BL pathogenesis, leading to loss of viral control and thereby contributing to BL oncogenesis [8]. While most research has focused on adaptive immunity, malaria-induced modifications in innate immune cells have been explored less extensively, yet these modifications may also play an important role in BL pathogenesis.

Neutrophils, the most abundant innate immune cell in human circulation [9,10], are rapidly recruited to sites of infection, where they deploy diverse effector functions, including the release of neutrophil extracellular traps (NETs), to limit viral replication and eliminate pathogens [11,12]. Beyond direct pathogen clearance, they have antigen-presenting capacity, allowing them to activate, shape, and regulate antiviral adaptive immune responses [13]. Neutrophils are increasingly being recognized as a heterogeneous population whose subsets dynamically adapt to physiological and pathological conditions [14]. However, extensive exploration of neutrophil biology in the context of chronic malaria has been limited. This is largely due to their short lifespan, limited survival after cryopreservation, and logistical challenges of performing real-time flow cytometric analyses in resource-limited settings typical of sub-Saharan Africa. As a result, the contribution of malaria-driven neutrophil reprogramming to BL pathogenesis remains largely unexplored.

Neutrophils exert their effector functions to clear blood-stage *Plasmodium falciparum (Pf)* parasites [15]; however, these functions are downregulated in children from malaria-endemic regions, making them susceptible to other opportunistic infections [16]. This downregulation has been linked to parasite-specific factors and proteins that inhibit neutrophil responses to proinflammatory signals [17], and reduced expression of neutrophil activation markers such as CD11a, CD11b, and CD18 has been observed in these children, suggesting malaria-induced impairment of neutrophil activation [18,19]. In malaria-endemic regions of western Kenya, where transmission is steady, and chronic *Pf*-exposure is common, children experience continuous immunological stimulation [20,21]. The long-term effects of repeated malaria exposure on neutrophil function are not well understood, and it is important to note that this is the same region where BL occurs, hypothesized to be a result of malaria-induced chronic immune dysregulation [8]. To date, there is limited information on how prolonged malaria-related changes in neutrophils might influence the broader immune environment and BL risk.

Inflammatory cues drive neutrophil plasticity [22–24], promoting the release of an immature subset from bone marrow and the retention of an aged subset in circulation [25], [26]. Persistent inflammation induces metabolic and transcriptional reprogramming in neutrophils, resulting in adaptations that can last for weeks to months [27]. As

a highly inflammatory disease, *Pf* malaria impairs both neutrophil effector functions [15–17,28], and activation in children [18]. In malaria endemic regions, where chronic *Pf* exposure is common, chronic immune stimulation may induce long-term transcriptional reprogramming of the neutrophil neutrotime, therefore altering circulating neutrophil subsets in ways that are not fully understood. It therefore remains unknown how chronic *Pf* malaria infection influences neutrophil subset dynamics and whether these alterations influence BL development and pathogenesis. Our study aims to fill this gap by ascertaining the effects of chronic *Pf*-malaria exposure on neutrophil subset profiles and evaluate their association with BL.

Our study characterized neutrophil subsets from the peripheral blood of malaria-exposed, malaria low-exposed, and BL children. Aged neutrophils are typically defined by the expression of the aging marker CD184 and loss of the adhesion molecule CD62L, which facilitates their migration to the site of inflammation. In addition to the conventional aged neutrophil subset ($CD62L^{neg}CD184^{pos}$), we identified an additional subset expressing both the aging marker CD184 and the adhesion molecule CD62L, and labelled it "atypical aged" ($CD62L^{pos}CD184^{pos}$). This subset was consistent in all our participants and, to our knowledge, has not been previously reported, suggesting increased heterogeneity in the aged neutrophil subset.

## Materials and methods

### Participants

We prospectively enrolled 18 participants within the age range of BL development, 4–9 years [29] from January 2024 to August 2024 at Moi Teaching and Referral Hospital (MTRH), Eldoret and processed the samples on site. This sample size was determined by the total number of suspected cases we were able to enroll at the MTRH, Eldoret, Kenya, during the timeframe of this study. The cancer diagnosis was made by flow cytometry, and the hospital pathologist confirmed the diagnosis by histology (hematoxylin and eosin tissue staining). Upon diagnosis, 13 of the enrolled suspected BL cases were confirmed, while 5 children were diagnosed with other types of cancers, such as Hodgkin's lymphomas, non-Hodgkin lymphomas, and sarcomas. To assess the impact of malaria on the BL neutrophil profiles, we collected samples from healthy children (cancer free, overall good health and without clinical symptoms of malaria such as fever for at least 48 hours prior sample collection) who were within a similar age range as the children with cancer (to limit age-associated bias): 19 healthy children residing in a malaria holoendemic area (malaria-exposed), and 11 healthy children residing in the low malaria transmission area (malaria low-exposed). The healthy children were enrolled based on the epidemiology of malaria where they reside. Post sampling, current and cumulative malaria exposure was confirmed by qPCR and serology, respectively. We also included Kenyan adult healthy controls. Based on our quality control criterions which includes quality and integrity of the samples (samples reaching the laboratory within 2 hours of blood draw were considered eligible for consistency in processing); quality of the flow cytometry data (a minimum of 450,000 cells acquired, a minimum of 98% of viability of these cells), only 3 adults were retained for the downstream analysis. Samples from malaria-exposed children were collected once at Ahero subcounty hospital in Kisumu County, a region known for high *Pf* malaria transmission, morbidity and mortality [30], and processed at KEMRI in May 2023 (single collection). Samples from the malaria low-exposed children and healthy adults were collected from Mosoriot hospital in Nandi County, a region with seasonal malaria transmission with a considerable year-to-year variation [31] known as a malaria hypoendemic area, in August 2024 (single collection).

### Ethics statement

Written informed consent was obtained from each adult participant and child's guardian. Ethical approvals were obtained from the Kenya Medical Research Institute (KEMRI) Scientific and Ethical Research Unit (approval number KEMRI/SERU/CGHR/05-06-386/4311), MTRH Institutional Research and Ethics Committee (IREC, approval number 0001911), and the University of Massachusetts Chan Medical School Institutional Review Board (IRB, approval number H00004587).

## Neutrophil Immunophenotyping

For the identification and characterization of neutrophils, 100 µL of whole blood was collected in lithium heparin tubes, and a nine-color panel was used to identify and characterize neutrophil subsets by flow cytometry. The antibodies in the panel are summarized in S1 Table. After staining, samples were fixed with 1X fixation and Permeabilization Solution containing 4.2% formaldehyde (BD Biosciences #Cat:51–2090KZ). After fixing, the samples were stored at 4°C and analyzed using the Beckman Coulter CytoFLEX Flow Cytometer the following day. The median of total live neutrophils, from the total participants, acquired by the flow cytometer is 351614 (145–66782). Compensation controls and FMOs were prepared for each run (*Full protocol detail in Supporting information*). Using FlowJo software, version 10.10.0, the compensation controls were used to correct for fluorescence spillovers, and the FMOs were used to set gates. A flow cytometry gating strategy was used for phenotyping neutrophils, and Boolean gating was used to identify neutrophil subsets (S1 Fig. **1A**). The CD62L FMO and gating strategy used to identify the aged and atypical aged subsets are shown in S1 Fig. **1B** and **1C**, respectively.

## Serological testing for AMA1 antibody levels

To confirm malaria exposure history, antibody levels against Apical Membrane Antigen 1 (AMA1, gift from Dr. Dutta, Walter Reed Army Institute of Research) were measured along with beads coupled to bovine serum albumin (BSA, Millipore Sigma) as internal control. Briefly, 50 µl of AMA1 and BSA conjugated beads (500 beads of each per well), suspended in assay buffer (PBS pH7.4 containing 0.1% BSA, 0.05% Tween-20, and 0.05% sodium azide) was added to each well of a flat bottom uncoated 96-well plate and a magnetic hand plate-washer was then used to wash the beads in plain assay buffer for 2 minutes. After discarding the supernatant, 50 µl of 1/100 diluted human plasma was added to the wells. A pool of plasma from healthy African adults, confirmed *Pf*-malaria seropositive, was used as a standard (2-fold dilution, S1 to S7) while the plasma from confirmed malaria-naive American adults was used as a negative control (dilution 1/100). The plate was incubated at room temperature for 2 hours. The beads were washed 3 times in the assay buffer and incubated for another 1 hour at room temperature in 50 µl of biotinylated anti-human IgG (BD, #Cat: 555785, dilution 1/1000). The washing of the beads was repeated three times, followed by a 15-minute incubation at room temperature with phycoerythrin-conjugated streptavidin (BD, #Cat: 554061, dilution 1/1000). After 15 minutes, the plate was washed for the final 3 times, and was read on a FlexMap 3D analyzer (Luminex). A minimum of 50 beads per analyte was acquired per well. The level of specific anti-AMA1 IgG from each of our studied groups was determined after subtraction of the BSA-beads median fluorescence intensity (MFI) to take into account possible non-specific binding.

## DNA extraction and malaria/EBV detection by qPCR

Genomic DNA was extracted from 100 µL of whole blood using the Zymo Quick DNA 96 kit (Zymo Research D3012) following the manufacturer's instructions. Briefly, blood samples were lysed in 400 µL Genomic Lysis Buffer before the mixture was transferred to a Silicon-A Plate on a Collection Plate and centrifuged at 3800 g (Eppendorf Centrifuge 5810) for 5 minutes. To obtain pure DNA, the extract was cleaned in 200 µL DNA Pre-Wash buffer followed by g-DNA Wash buffer, spinning each wash at 3800g for 5 minutes and discarding the supernatants. The DNA was eluted in 30 µL of the DNA Elution Buffer and stored at -20°C for downstream analysis. The quality of the DNA product was confirmed using spectrophotometry (Thermo Scientific Nanodrop 2000) at A260/A280 and A260/A230.

For malaria parasite detection, we utilized an ultrasensitive quantitative real-time polymerase chain reaction (qPCR) based on the varATS region using forward primer (5'-CCCATACACAACCAAYTGGA-3'), reverse primer (5'-TTCGCA CATATCTCTATGTCTATCT-3') and probe sequences (6-FAM-TRTTCCATAAATGGT-NFQ-MGB), originally described in [32]. To prepare 12.5 µL of reaction mix per well, the following reagents were added: 4.75 µL of DNA template, 6.25 µL of TaqMan Gene Expression Mastermix (Thermo fisher 43690116), 0.5 µL of Forward primer at a concentration of 20 µM (IDT), 0.5 µL of

the Reverse primer at a concentration of 20 µM (IDT) and 0.5 µL of Molecular grade Water (Thermo Fisher 10977035). The 12.5 µL reaction mix was loaded into a 0.1 mL MicroAmp Fast 96-Well Reaction plate (Thermo Fisher 4346907), sealed with MicroAmp Optical Adhesive Film (Thermo Fisher 431197), and loaded on QuantStudio 6 pro qPCR machine (Thermo Fisher A43180). The qPCR run protocol was set for 2 minutes pre-incubation at 50°C, 10 minutes initial denaturation at 95°C, and 45 cycles of 15 seconds denaturation at 95°C followed by 1 minute annealing and elongation steps at 55°C. The qPCR output was analysed using Design and Analysis Software version 2.6 QuantStudio 6/7 Pro Systems (Thermo Fisher). For quality control, the samples were run alongside replicates of positive and negative controls. We applied a standard curve analysis to determine downstream parasite densities using 10-fold serial dilution of 3D7 strain DNA as *Pf* positive control (10,000, 1,000, 100, 10, 1, and 0.1 copies per µL). The positivity threshold was set-up at Ct value of 20 cycles.

To detect EBV, we used the following primers and probe targeting BALF5 (forward: 5′-CGGAAGCCCTCTGGACTTC-3′, reverse: 5′-CCCTGTTTATCCGATGGAATG-3′, probe: 5′-FAM-TGTACACGCACGAGAAATGCGCCT-BHQ1–3′). We also quantified human beta-actin as the reference in this assay as described in previous work [33–35]. Reactions were run in a 12 µl total volume containing 6.0 µl of 2×iQ Supermix (Bio-Rad), PCR-grade water (2.5 µl), and 2.0 µl of template DNA. Final primer/probe concentrations were 500 nM each for EBV BALF5 forward and reverse primers (0.6 µl each of 10 µM stocks), 100 nM for the EBV probe (0.12 µl of a 10 µM stock), and 50 nM each for beta-actin forward primer, reverse primer, and probe (0.06 µl each of 10 µM stocks). qPCR was performed on a Bio-Rad CFX96 Real-Time System with a C1000 Thermal Cycler base using the following cycling conditions: 95°C for 3 minutes, followed by 40 cycles of 95°C for 10 seconds and 62.5°C for 30 seconds, with fluorescence acquisition at the end of each 62.5°C step. Quantification was carried out using standard curves generated from Namalwa genomic DNA (ATCC CRL-1432, which contains two integrated EBV copies per cell). For EBV, standards were prepared as six serial 5-fold dilutions, ranging from 10^5–10^1 copies per reaction, and each was run in duplicate. The EBV copy number in each reaction was determined by interpolating from the EBV standard curve and normalized to cellular equivalents based on beta-actin quantification, yielding EBV copies per cell.

## Statistical analysis

The flow cytometry data was analyzed using FlowJo version 10.10.0. Statistical analysis was done in R software version 4.2.1. Two BL Samples with low cell count and viability of less than 80% were excluded from analysis bringing the BL sample size from 13 to 11. The data was not normally distributed, so the Mann-Whitney test was used to compare subset frequency between the two groups. For comparisons involving more than two groups, the Kruskal-Wallis test was applied to detect overall differences, followed by Dunn's post hoc test, allowing for Dunn's multiple comparison correction. Since the data was not uniformly distributed, the Spearman correlation was used. A predetermined $p$-value < 0.05 was considered statistically significant.

## Results

### Characteristics of the study participants

As expected, children with cancer had significantly higher Absolute Neutrophil Count (ANC, $p = 0.0001$, $p = 0.02$) and White Blood Cell Count (WBC, $p = 0.01$, $p = 0.03$) compared to healthy control children. Children with other childhood cancers showed significantly higher ANC counts compared to BL children ($p = 0.04$, S2 Table). ANC and WBC were comparable between malaria-exposed and malaria-low-exposed healthy children (S2 Table). We found that 68% of the malaria-exposed healthy controls had asymptomatic malaria infections, while none of the malaria low-exposed children or adults had malaria parasitemia by qPCR (S2 Table and S3 Table, respectively). As expected, malaria-exposed children had significantly higher specific anti-AMA1 IgG antibody levels than malaria low-exposed children: (median MFI 72126, range [384–217702] vs median MFI 0, range [0–2541]; $p = 0.0001$, S2 Table). A similar observation was made between BL and malaria low-exposed children ($p = 0.005$, S2 Table), confirming the malaria cumulative exposure of our BL and

malaria-exposed group of children. There was no significant difference in specific anti-AMA1 IgG antibody levels between malaria-exposed children and children with BL (median MFI 72126, range [384–217702] vs median MFI 2495, range [0–171811]; *p = 0.10,* S2 Table), as well as between children with BL and children with other cancers (median MFI 2495, range [0–171811] vs median MFI 197, range [0–153896]; *p = 0.27,* S2 Table). Malaria-exposed children had significantly higher parasitaemia compared to malaria low-exposed children, and children with BL, (*p = 0.0001, p = 0.001,* respectively, S2 Table). There was no significant difference in parasitaemia between malaria low-exposed children vs children with BL as well as between children with BL vs children with other childhood cancers (*p = 0.11, p = 0.88,* respectively, S2 Table). Children with BL had significantly higher EBV viral loads compared to children with other childhood cancers (*p = 0.01,* S2 Table). The demographic data for healthy Kenyan adults is summarized in S3 Table.

### Malaria shifts "mature" neutrophils subsets towards "aged" phenotype

We characterized and compared the frequency of neutrophil subsets in malaria-exposed (ME) and malaria low-exposed (MLE) children and found that MLE children had significantly higher frequencies of mature active (CD11b$^{pos}$) neutrophils compared to ME (MLE median 31%, range [16.4%–56.3%] vs. ME median 2.3%, range [0%–4.97%]; *p = 0.0035,* Fig 1A). Similar observation was made regarding the frequencies of mature inactive (CD11b$^{neg}$) neutrophil (median 0% vs. median 0.08%, range [0–0.37]; *p < 0.0001,* Fig 1B). No statistical difference was found in regards to immature neutrophils between MLE and ME (Fig 1C), however, we noticed significantly high frequencies of atypical aged and aged neutrophils in ME children compared to MLE ones with a median of 35.8% atypical aged neutrophils in ME, range [11.7%–49.3%] vs. median 0.02% in MLE, range [0.02%–0.64%] *p < 0.0001,* Fig 1D; and a median of 9.35% aged neutrophils in ME, (range [3.2%–34.6%]) vs. a median of 0.3% in MLE (range [0.15%–0.78%]), *p < 0.0001,* Fig 1E). Together this suggests that malaria exposure is driving the expansion of the aged neutrophils.

### Immature neutrophil is the main subset in the peripheral blood of children with BL

We then compared neutrophil subsets in healthy and BL children. Children with BL showed dramatically lower frequency of the mature active subset compared to healthy children with a median of 0.02% of mature active neutrophils in BL (range [0%−0.23%]) compared to a median of 2.3% in ME (range [0%−4.97%]); *p = 0.0194*) and a median of 31% in MLE (range [16.4%–56.3%]); *p < 0.0001,* Fig 1A). Following the same trends, children with BL had no mature inactive neutrophils leading to a strong statistical difference with their frequencies in MLE (*p < 0.0001,* Fig 1B). Interestingly, children with BL had significantly higher frequency of the immature neutrophil subset compared to healthy ME children (median 67.5%, range [8.16%−99.6%] vs median 12%, range [4.29%−59.8%]; *p = 0.006,* Fig 1C). Finally, healthy ME children had higher frequency of aged and atypical neutrophils compared to children with BL and this difference was statistically significant for the atypical aged neutrophils (median 35.8%, range [11.7%−49.3%] vs median 3.59%, range [0.07%−35.9%]; *p = 0.023,* Fig 1D). No statistical difference was found in regards to aged neutrophils between healthy children and BL (Fig 1E). Of note, we did not observe any differences either in the neutrophil subset profiles between BL participants who had asymptomatic malaria and the ones who did not (S2 Fig).

Using Spearman correlation, we determined the relationship between neutrophil subsets in ME and BL children. A significant positive correlation was observed in the immature subset (*rs = 0.70, p < 0.0001*) whereas negative correlations were observed regarding the mature active (*rs = −0.63, p = 0.0002*), atypical aged (*rs = −0.70, p < 0.0001*), aged (*rs = −0.32, p = 0.09*), and mature inactive subsets (*rs = −0.14, p < 0.46*) (Table 1).

### Cumulative exposure to malaria during childhood leads to reduced abundance of mature active and inactive neutrophils, reaching levels similar to those observed in adults

There was no difference in the mature active subset frequency between adults and ME, however, MLE children had significantly higher frequency than both the adults and ME children (*p = 0.0004, p = 0.0002,* respectively Fig 2A). Similarly,

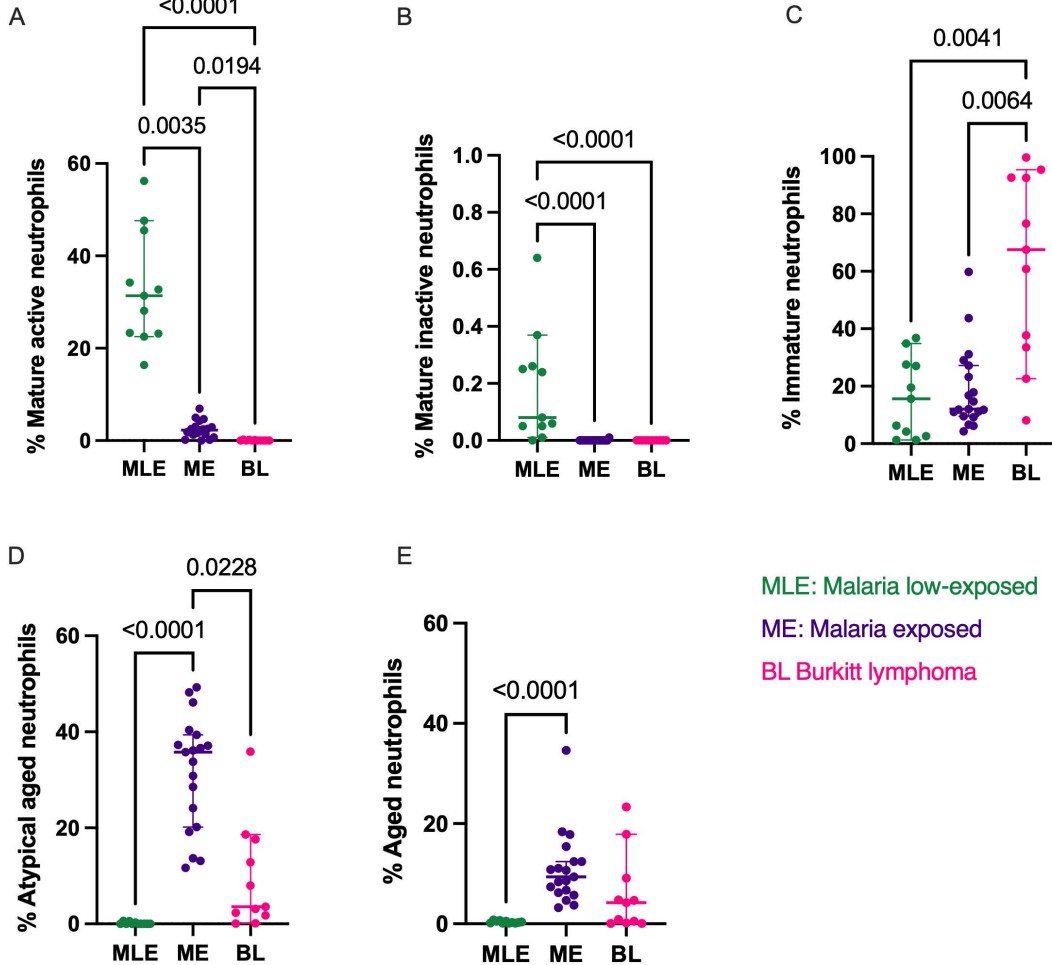

**Fig 1. Comparison of neutrophil subset frequency between healthy and BL children. A.** Mature active (CD11b$^{pos}$CD62L$^{pos}$CD182$^{pos}$CD184$^{neg}$), **B.** Mature-inactive (CD11b$^{neg}$CD62L$^{pos}$CD182$^{pos}$ CD184$^{neg}$), **C.** Immature (CD10$^{neg}$), **D.** Aged (CD62L$^{neg}$CD184$^{pos}$), and **E.** Atypical aged (CD62L$^{pos}$CD184$^{pos}$) neutrophilsfrom malaria low-exposed (MLE, green, n = 11), malaria-exposed (ME, purple, n = 19) and children with BL (pink, n = 11). The bars represent median and 95% confidence intervals. Kruskal-Wallis test with Dunn's correction for multiple comparisons was used, *p*-values <0.05 are indicated in the figure.

**Table 1. Spearman correlation of neutrophil subsets between malaria-exposed children and BL.** Neutrophil subsets were defined as: Mature active (CD11b$^{pos}$CD62L$^{pos}$CD182$^{pos}$CD184$^{neg}$), Atypical Aged (CD62L$^{pos}$CD184$^{pos}$), Aged (CD62L$^{neg}$CD184$^{pos}$), Immature (CD10$^{neg}$), and Mature Inactive (CD11b$^{neg}$CD62L$^{pos}$CD182$^{pos}$CD184$^{neg}$). Correlation coefficients (rs) and *p*-values were calculated using the Spearman rank test.

| Subsets | Mature Active | Atypical Aged | Aged | Immature | Mature Inactive |
|---|---|---|---|---|---|
| Spearman's correlation(rs) | −0.63 | −0.70 | −0.32 | **0.70** | −0.14 |
| *P–value* | p = 0.0002 | p < 0.0001 | p = 0.09 | **p < 0.0001** | p = 0.46 |
| *95% CI for P* | −0.81 to-0.35 | −0.84 to-0.45 | −0.61 to 0.05 | **0.45 to 0.84** | −0.48 to 0.23 |
| *Sample size* | n = 30 | n = 30 | n = 30 | n = 30 | n = 30 |

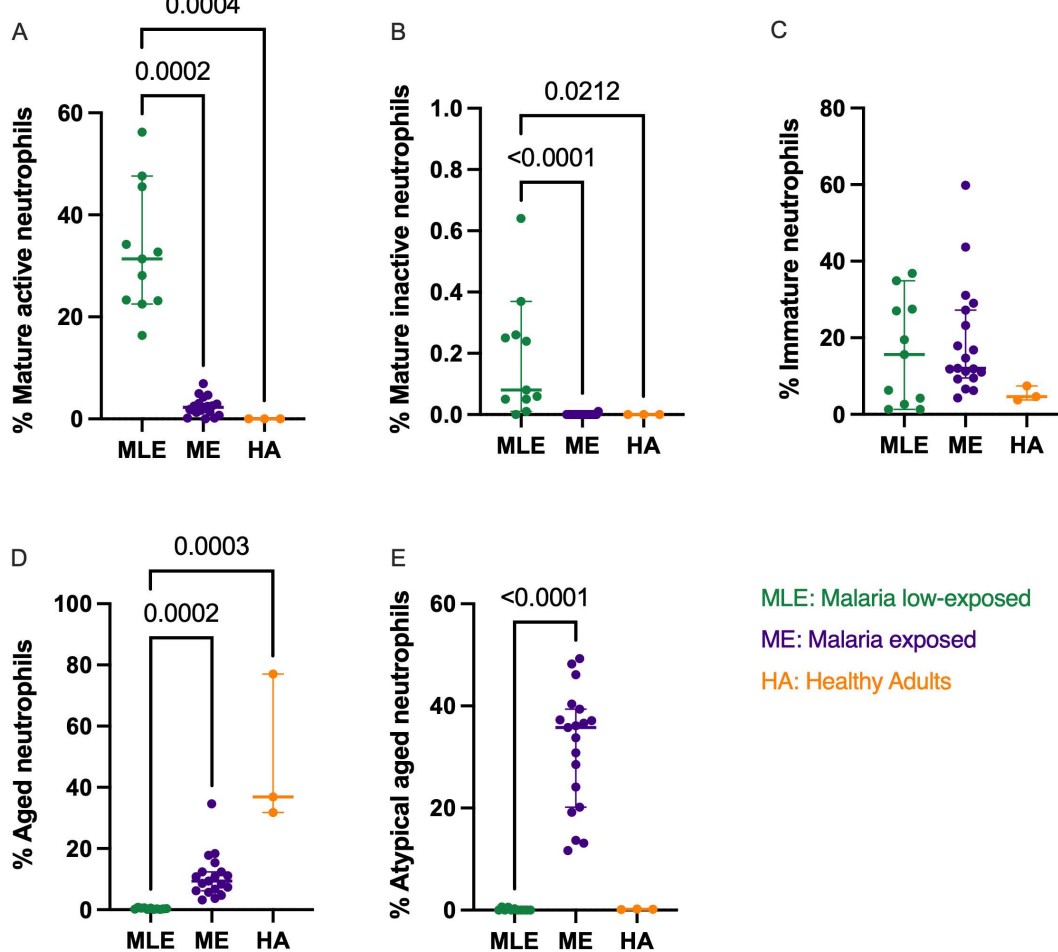

**Fig 2. Comparison of neutrophil subset frequency in adults, malaria-exposed children, and malaria low-exposed children. A.** Mature active (CD11b$^{pos}$CD62L$^{pos}$CD182$^{pos}$CD184$^{neg}$), **B**. Mature-inactive (CD11b$^{neg}$CD62L$^{pos}$CD182$^{pos}$ CD184$^{neg}$), **C.** Immature (CD10$^{neg}$), **D.** Aged (CD62L$^{neg}$CD184$^{pos}$), and **E.** Atypical aged (CD62L$^{pos}$CD184$^{pos}$) neutrophilsfrom malaria low-exposed (MLE, green, n = 11); malaria-exposed children (ME, purple, n = 19); and healthy adults (HA, orange, n = 3). The bars represent median and 95% confidence intervals. Kruskal-Wallis test with Dunn's correction for multiple comparisons was used, *p*-values<0.005 are indicated in the plots.

only MLE children had significantly higher frequency of mature inactive neutrophils compared to both adults and ME children (*p = 0.0212, p < 0.0001*, respectively Fig 2B). No differences were observed across groups in regards to the immature neutrophil subset (Fig 2C), however, both adults and ME children had significantly higher frequency of the aged neutrophils compared to the MLE children (*p = 0.0003, p = 0.0002*, respectively, Fig 2D) while no difference was observed between adults and ME. Finally, atypical aged neutrophil subset was found significantly more abundant in ME compared to MLE children (*p < 0.0001*, Fig 2E).

### Children with non-BL cancers have a higher frequency of the atypical aged neutrophil subset

To assess whether the observed neutrophil profiles were unique to BL, we compared them to children with other cancers (OC). No statistical differences were observed in the frequencies of the mature active, mature inactive, immature, and aged neutrophil subsets (**Fig. 3A**/**B/C/D**). The only difference observed was for the atypical aged subset which was higher in the

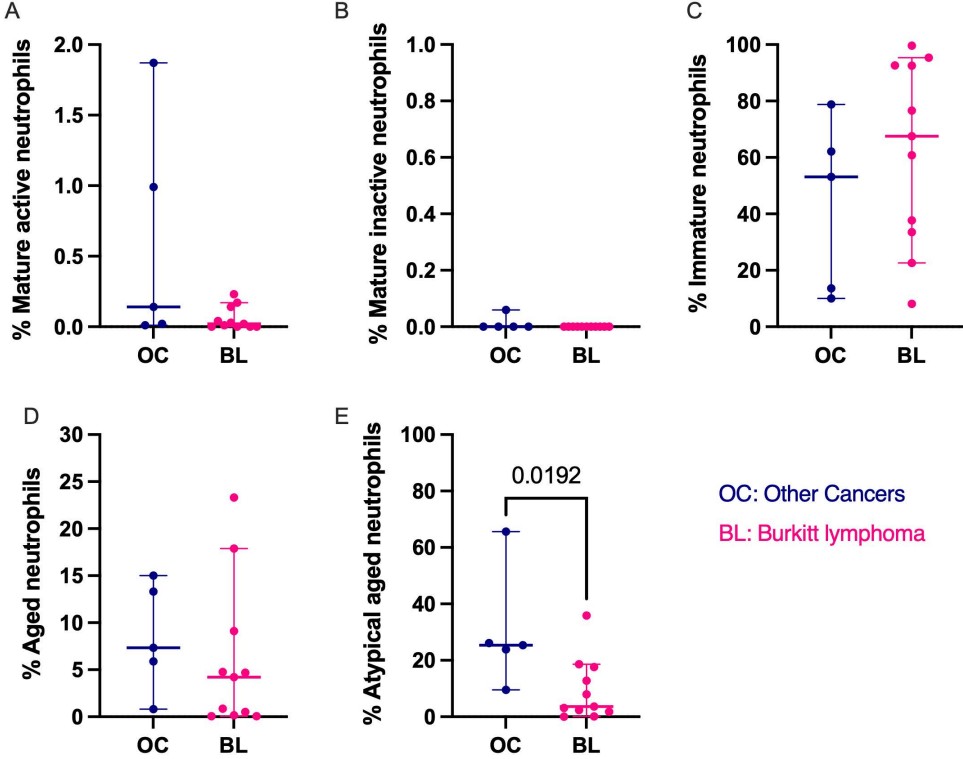

**Fig 3. Comparison of neutrophil subset frequency between children with eBL and children with non-BL cancers. A.** Mature active (CD11b$^{pos}$CD62L$^{pos}$CD182$^{pos}$CD184$^{neg}$), **B**. Mature-inactive (CD11b$^{neg}$CD62L$^{pos}$CD182$^{pos}$ CD184$^{neg}$), **C.** Immature (CD10$^{neg}$), **D.** Aged (CD62L$^{neg}$CD184$^{pos}$), and **E.** Atypical aged (CD62L$^{pos}$CD184$^{pos}$) neutrophilsfrom children with other childhood cancers (OC, blue, n = 5); and children with BL (pink, n = 11). The bars represent Median and 95% confidence intervals. Mann-Whitney non-parametric test was used. *P*-values<0.05 are indicated in the plots.

OC compared to BL patients (median of 25.4% atypical aged in OC, range [9.54–65.6] vs median of 3.59% in BL, range [0.07%−35.9%]; *p*=0.019, Fig 3E). These findings suggest that there are immunological changes unique to non-BL cancers.

## Discussion

Chronic *Pf*-malaria exposure appears to alter proportions of neutrophil subsets by significantly increasing the frequency of the aged and atypical aged subsets, while significantly reducing the frequency of both the mature active and mature inactive neutrophil subsets. This shift in neutrophil subsets was also observed in adults, however, chronic malaria exposure drives a shift towards an atypical aged neutrophil phenotype in malaria-exposed children whereas, as expected, conventional aged neutrophils were the most abundant subset in adults. Additionally, we found that the frequencies of the immature subset in malaria-exposed children and BL children showed a significant correlation.

Chronic malaria exposure has been linked to defects in immunosurveillance, contributing to BL pathogenesis [4,36]. Mounting evidence suggests that repeated clinical episodes of *Pf* malaria result in substantial modification of the host immune system, significantly altering the phenotype and function of several immune cell populations [37,38]. To understand the effect of repeated malaria exposure on neutrophil subsets, we characterized and compared neutrophil subsets between malaria-exposed and malaria low-exposed children. The aged neutrophil subset (CD62L$^{neg}$CD184$^{pos}$) and atypical-aged (CD62L$^{pos}$CD184$^{pos}$) frequencies were significantly higher in malaria-exposed children. At homeostasis, aged neutrophils exit circulation and migrate to the bone marrow for elimination [39]. However, during acute inflammation, instead of returning to the bone marrow, they rapidly migrate to the inflamed site, leading to elevated levels of aged neutrophils in circulation

[26]. Given that 62% of our malaria-exposed healthy controls had asymptomatic malaria, which is a chronic, low-grade parasite-driven inflammation, our results suggest that malaria, just like other inflammatory diseases, may be driving a shift in neutrophil subsets, and this may be leading to the accumulation of aged neutrophil subsets in circulation.

These malaria-exposed children also had significantly low frequencies of both mature active (CD11b$^{pos}$ CD62L$^{pos}$ CD182$^{pos}$ CD184$^{neg}$) and mature-inactive (CD11b$^{neg}$CD62L$^{pos}$ CD182$^{pos}$ CD184$^{neg}$) subsets. Inflammatory signals and elevated cytokine levels have been shown to modulate the expression of CD182, a critical chemokine receptor that regulates the egress of mature neutrophils from the bone marrow to inflammatory sites [40]. This modulation has been linked to a reduced frequency of mature neutrophils in circulation in various inflammatory diseases, such as sepsis [40–42], therefore suggesting that the parasite-induced chronic inflammation could be driving the observed low frequency of the mature subsets in malaria-exposed children. In addition to the reduced expression of neutrophil activation marker CD11b previously reported in children with malaria [18], our findings confirm that, similar to other inflammatory diseases, chronic malaria exposure disrupts neutrophil subset homeostasis, leading to the accumulation of aged neutrophil subsets and a concomitant reduction in mature subsets in circulation.

In addition to the aged neutrophil subset (CD62L$^{neg}$CD184$^{pos}$) reviewed in [43], we identified a distinct aged subset expressing CD62L in our participants, which, to our knowledge, has not been previously reported. This finding is particularly intriguing given that aged neutrophils are typically characterized by downregulation of CD62L [44]. CD62L, or L-selectin, is a cell adhesion molecule that mediates the multistep neutrophil trans endothelial migration process [45]. As neutrophils age, there is downregulation of CD62L and upregulation of CD184 to facilitate their homing to the bone marrow in a rhythmic fashion for efferocytic clearance [46]. While our data demonstrate phenotypic heterogeneity within the aged neutrophil compartment, the underlying dynamics and functional properties of these subsets were not assessed. Our observation, therefore, raises the possibility of additional heterogeneity and functional differences within the aged neutrophil compartment that could be relevant to infection-driven and cancer-related immune responses. Further studies are needed to characterize the function and origin of this atypical subset.

Cancer studies have shown that tumors trigger emergency granulopoiesis by producing elevated levels of granulocyte colony-stimulating factor (G-CSF) [47,48]. This phenomenon was described as a tumor-induced mechanism that can lead to excessive recruitment of immature neutrophil subsets into the peripheral blood, facilitating immunosuppression and promoting metastasis [49]. In our study, we found significantly higher frequency of immature (CD10$^{neg}$) neutrophil subsets in children with BL compared to the malaria-exposed children, suggesting that BL, just like other cancers, induces excessive recruitment of immature neutrophils as a mechanism of immune evasion and progression. Moreover, a strong positive correlation we found regarding the immature neutrophil subset between malaria-exposed and BL children suggests a similar expansion pattern of this subset in both groups. Despite the established link between cancer and immature neutrophils, our observations suggest that chronic or repeated malaria exposure may also be contributing to the elevated presence of immature neutrophils, pointing to a possible immunological bridge between malaria and BL pathogenesis.

We observed some similarities in the neutrophil subset frequencies between malaria-exposed children and healthy adults for whom a shift from mature to aged neutrophils occurred. Indeed, adults had a significantly high frequency of the aged (CD62L$^{neg}$CD184$^{pos}$) while atypical aged (CD62L$^{pos}$CD184$^{pos}$) subset was more abundant in malaria-exposed children. Aging has been shown to directly affect neutrophil homeostasis and functions [50,51]. Elderly individuals over the age of 65 have shown accumulation of the aged (CD62L$^{neg}$CD184$^{pos}$) neutrophil subsets in their circulation, playing a role in age-related pathologies such as periodontitis [52]. This subset has been shown to have altered migration patterns, enhanced activation, and increased production of reactive oxygen species (ROS), which can contribute to inflammation and tissue damage [53]. Our findings align with earlier studies showing that aging and inflammatory diseases, such as malaria disrupts neutrophil subset balance, leading to a high frequency of aged neutrophils in circulation [26]. We also observed a significantly higher frequency of the atypical aged subset (CD62L$^{pos}$CD184$^{pos}$) in malaria-exposed children. The elevated levels of both aged (CD62L$^{neg}$CD184$^{pos}$) and atypical aged (CD62L$^{pos}$CD184$^{pos}$) neutrophils in malaria-exposed children suggest that chronic malaria exposure may drive a shift toward an aged neutrophil phenotype. However, functional assays comparing these

subsets in adults and malaria-exposed children are needed to determine whether they share similar immunological roles. We also observed that the healthy adults and malaria-exposed children have a significantly low frequency of both the mature active (CD11b$^{pos}$CD62L$^{pos}$CD182$^{pos}$CD184$^{neg}$) and mature inactive (CD11b$^{neg}$CD62L$^{pos}$ CD182$^{pos}$CD184$^{neg}$) neutrophil subsets in circulation. Mature neutrophils circulate in the blood in an inactive form [54], and when they encounter pathogens or inflammatory signals, they become activated [55]. The mature active neutrophils are part of the body's first defence against infections and inflammations [56]. Reduced expression of the neutrophil activation marker, CD11b, has been observed in children with severe malaria [18,19]. This, therefore, suggests that the low frequency of the mature neutrophil subsets observed reflects a malaria-induced impairment of neutrophil activation in malaria-exposed children. In addition, a significant decline in the release of neutrophils from the bone marrow into the bloodstream in older individuals has been observed [52], and this could be contributing to the low frequency of the mature active neutrophil subsets we observed in adults in our study. We acknowledge the small sample size of our healthy adult group, limiting the interpretation of our results.

To assess whether the observed neutrophil subsets were unique to BL and whether their dysregulation was specific to BL or a common feature in childhood cancers, we characterized neutrophil subsets in children with BL and those with other childhood cancers. Atypical aged neutrophils (CD62$^{pos}$CD184$^{pos}$) were found to be of higher frequencies in non-BL cancers as opposed to BL, suggesting that neutrophil dysregulation may not be uniform in BL and other childhood cancers. In neutrophils, CD62L is a crucial cell adhesion molecule that facilitates their migration to the site of inflammation [57]. Previous studies in cancer have shown that tumors exploit the aged neutrophils to drive inflammation and promote metastasis [58]. Therefore, the significantly higher frequency of the atypical aged neutrophils (CD62L$^{pos}$CD184$^{pos}$) observed in our study suggests that non-BL cancers could be inducing the accumulation of aged neutrophils that still express CD62L. However, extensive work is required to functionally distinguish between atypical aged neutrophils still expressing CD62L and aged neutrophils that no longer express CD62L.

## Conclusion

In conclusion, our work demonstrates that chronic malaria exposure reshapes neutrophil composition by increasing the frequency of aged neutrophils while reducing mature neutrophils. A confirmation cohort occurring on a longer period of time to increase the number of enrolled BL cases is needed to strengthen our results. However, the association between immature neutrophils and BL found in our study provides insights into how these malaria-driven immune alterations may potentially contribute to BL immuno-pathology.

## Supporting information

**S1 Fig. Gating strategy. (A)** Flow cytometry gating strategy used to phenotype neutrophils. **(B)** CD62L gating based on FMO for malaria-exposed and malaria low-exposed children, adults, children with BL or other cancer. **(C)** Gating strategy used to identify atypical aged neutrophil subset.
(TIFF)

**S2 Fig. Neutrophil subset profiles between BL participants with and without *Pf* parasitemia.**
(TIFF)

**S1 Table. Neutrophil flow cytometry antibody panel.**
(DOCX)

**S2 Table. Demographics and Characteristics of the Study Participants.**
(DOCX)

**S3 Table. Demographics and Characteristics of the Adult Participants.**
(DOCX)

**S1 File. Malaria and EBV load qPCR raw data.**
(XLSX)

**S2 File. Neutrophils subsets raw data.**
(XLSX)

**S3 File. AMA1 serology raw data.**
(XLSX)

**S4 File. Visual Abstract Neutrophils manuscript.**
(JPEG)

## Acknowledgments

The authors want to thank the KEMRI-UMASS and AMPATH Reference Lab teams at Moi Teaching and Referral Hospital for their incredible work in disease diagnosis and patient care. We also want to thank our study participants for consenting to be part of this study and for giving their samples for this work to be accomplished.

## Author contributions

**Conceptualization:** Sharon Akinyi Okoth, Cyrus Ayieko, Ann W. Kinyua, Catherine S. Forconi.

**Data curation:** Sharon Akinyi Okoth.

**Formal analysis:** Sharon Akinyi Okoth, Titus K. Maina, Eddy Agwati.

**Funding acquisition:** Ann M. Moormann.

**Investigation:** Sharon Akinyi Okoth, Cliff I. Oduor, Zachary Racenet, Viriato M'Bana.

**Methodology:** Sharon Akinyi Okoth, Eddy Agwati, Ann W. Kinyua, Catherine S. Forconi.

**Project administration:** Ronald K. Tonui, Cliff I. Oduor, Festus M. Njuguna, Kibet K. Keitany, Ann M. Moormann.

**Resources:** Ronald K. Tonui, Cliff I. Oduor, Festus M. Njuguna, Kibet K. Keitany, Daniel Chepsiror, Ann M. Moormann.

**Supervision:** Cyrus Ayieko, Ann W. Kinyua, Catherine S. Forconi.

**Validation:** Ann W. Kinyua, Catherine S. Forconi.

**Visualization:** Sharon Akinyi Okoth, Titus K. Maina, Catherine S. Forconi.

**Writing – original draft:** Sharon Akinyi Okoth.

**Writing – review & editing:** Sharon Akinyi Okoth, Ronald K. Tonui, Titus K. Maina, Eddy Agwati, Cliff I. Oduor, Zachary Racenet, Viriato M'Bana, Cyrus Ayieko, Ann M. Moormann, Ann W. Kinyua, Catherine S. Forconi.

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
