## [Decision Letter · Decision Letter 0]

26 Jan 2026

PONE-D-25-56348The Impact of Malaria-Induced Neutrophil Subset Shift and a Link to Burkitt LymphomaPLOS One

Dear Dr. Forconi,

Thank you for submitting your manuscript to PLOS ONE. Upon careful assessment, we conclude that while the manuscript demonstrates merit, it does not currently meet all of the criteria for publication. We invite you to submit a revised manuscript that responds to the issues raised during peer review. Please submit your revised manuscript by Mar 12 2026 11:59PM. If you will need more time than this to complete your revisions, please reply to this message or contact the journal office at plosone@plos.org. Please include the following items when submitting your revised manuscript:

We look forward to receiving your revised manuscript.

Kind regards,

Srinivasa Reddy Bonam

Academic Editor

PLOS One

Journal Requirements:

[NIH R01 CA189806 (Moormann).].

3. Thank you for stating the following in your manuscript:

[This study was funded by the NIH R01 CA189806 (Moormann).]

[NIH R01 CA189806 (Moormann).]

5. Please upload a new copy of Figure S1 as the detail is not clear. Please follow the link for more information:  https://journals.plos.org/plosone/s/figures

Reviewers' comments:

Reviewer's Responses to Questions

**Comments to the Author**

1. Is the manuscript technically sound, and do the data support the conclusions?

Reviewer #1: Partly

Reviewer #2: Yes

Reviewer #3: Yes

Reviewer #4: Partly

2. Has the statistical analysis been performed appropriately and rigorously? 

Reviewer #1: No

Reviewer #2: Yes

Reviewer #3: Yes

Reviewer #4: Yes

3. Have the authors made all data underlying the findings in their manuscript fully available?

Reviewer #1: Yes

Reviewer #2: Yes

Reviewer #3: Yes

Reviewer #4: Yes

4. Is the manuscript presented in an intelligible fashion and written in standard English?

Reviewer #1: Yes

Reviewer #2: Yes

Reviewer #3: Yes

Reviewer #4: No

5. Review Comments to the Author

Reviewer #1: Major comments

1. Individuals residing in malaria-prone areas whether seasonal, epidemic-prone, or endemic cannot be appropriately classified as malaria-naïve. The presence of positive malaria results (see S2 Table) further invalidates the use of the term “malaria-naïve” for this group. The authors should therefore adopt an alternative and more accurate terminology to describe these participants.

2. The term “healthy group” should be explicitly defined in the Methods section to avoid ambiguity for example, whether “healthy” refers to the absence of cancer, malaria infection, or other clinical conditions.

3. Include the number of malaria positive cases by PCR in malaria naïve participants and the 3 adults in text within the result section.

4. The criteria used to select the three adult participants are unclear. The authors should justify why only three adults were included and explain why this number differs from the fixed number of children selected.

5. Given the very small number of adults included (n = 3), the data are insufficient to support the conclusions drawn from this group….“Malaria-exposed children also had neutrophil profiles that closely resembled those seen in the adults.” Also, among the adult samples, AMA1A (MFI) was 31,108 [104-5977] indicating prior interaction with Plasmodium. If this was detected in one of the three adult samples, it could represent a substantial proportion relative to the small sample size and may differ considerably from observations made in larger cohorts, thus misleading.

Reviewer #2: The manuscript titled ‘The Impact of Malaria-Induced Neutrophil Subset Shift and a Link to Burkitt Lymphoma’ presents a strong and interesting study, but the following concerns should be addressed before publication.

Major:

1. The authors should clarify the difference between ‘mature’ and ‘mature inactive’ neutrophils, as both populations appear to share the same surface marker expression in the Discussion (lines 286–287) and in the legends of Figures 1 and 3.

2. Neutrophil subsets are defined using surface marker expression; however, these markers do not explain functional heterogeneity or suppressive activity. Consequently, the immunosuppressive or tumor-promoting roles linked to the expanded immature and aged neutrophil subsets remain speculative.

3. Statements made in lines 311-313 should be clearly supported by the corresponding dataset

4. For lines 149-151, the authors should provide the Ct values and define the thresholds and criteria used for downstream parasite quantification. Also add the primer seq in supplementary material.

5. The authors should clarify whether malaria-exposed children (n = 19) were defined by residence or by lab data, and specify the proportion with active, treated, and asymptomatic malaria at the time of sampling.

6. The authors infer that a malaria-driven expansion of immature neutrophils contributes to BL immunopathology; however, no supporting BL-specific data are provided. Inclusion of BL-associated protein or transcriptional markers would substantiate this link.

Minor:

a. The authors should include the ethics approval ID numbers for all approving institutes.

b. Sentence referring to the graphical abstract should be removed from lines 215-216.

c. Figure S1 resolution should be improved, and need to include all relevant dot plot data to the Supplementary Material.

Reviewer #3: The authors have presented a well written and detailed account on understanding the shift in neutrophil profiles observed during malaria-induced immunopathy, associated with aggressive Burkitt lymphoma. The data is precise and the manuscript is easy to follow.

Reviewer #4: Overall: The manuscript’s primary strength is describing malaria-associated neutrophil phenotypes and a novel use-case; however, the extension to a BL pathogenesis ‘bridge’ is insufficiently supported by the presented data and should be framed as hypothesis-generating or more exploratory, specification of this might strengthen ties to the conclusions. The methods and gating strategies are clearly elucidated with pertinent details for reproducibility. Detailed critique is below:

Introduction: would benefit from a clearer explanation of what is known about neutrophils in malaria, especially differences between acute infection vs chronic/repeated exposure, and why these distinctions matter for interpreting potential links to BL. The authors should state explicitly what gap their study fills-specially as the paper is being presenting as bridging the link between malaria-neutrophils and BL.

Definitive wording: words such as "defects" in the introduction are too strong, precise immunological mechanisms remain incomplete and not fully mapped, impairment might be a better word to address these.

Study design: unclear if the malaria exposure was quantified for BL participants and how it was incorporated into analyses; exposure differences could substantially affect interpretation. The manuscript highlights BL, was EBV status/viral load (and tumor EBV status where applicable) was assessed? It may be difficult to distinguish malaria-associated effects from cancer/EBV-driven immune remodelling in this cross-sectional design. Separating BL and non-BL associated effects, malaria vs cancer driven effects and so on might add value to the main interplay the paper wants to address.

Statistics: small sample size has been stated as a limitation in the discussion, authors might want to exercise caution on the "correlation vs causation" front.

Writing and Grammar: The manuscript will benefit from improved flow and language check to avoid making very strong claims that may not have fully supporting data, suggestive wording might be better. Line 347 to 349 is incomplete.

6. PLOS authors have the option to publish the peer review history of their article (what does this mean?). If published, this will include your full peer review and any attached files.

Reviewer #1: No

Reviewer #2: No

Reviewer #3: No

Reviewer #4: No

---

## [Author Response · Author response to Decision Letter 1]

19 Mar 2026

The responses to all reviewer comments have been submitted as an attachment file

---

## [Decision Letter · Decision Letter 1]

20 Apr 2026

The Impact of Malaria-Induced Neutrophil Subset Shift and a Link to Burkitt Lymphoma

PONE-D-25-56348R1

Dear Dr. Catherine S. Forconi,

We’re pleased to inform you that your manuscript has been judged scientifically suitable for publication and will be formally accepted for publication once it meets all outstanding technical requirements.

Kind regards,

Srinivasa Reddy Bonam

Academic Editor

PLOS One

Reviewers' comments:

Reviewer's Responses to Questions

**Comments to the Author**

1. If the authors have adequately addressed your comments raised in a previous round of review and you feel that this manuscript is now acceptable for publication, you may indicate that here to bypass the “Comments to the Author” section, enter your conflict of interest statement in the “Confidential to Editor” section, and submit your "Accept" recommendation.

Reviewer #1: All comments have been addressed

Reviewer #2: All comments have been addressed

2. Is the manuscript technically sound, and do the data support the conclusions?

Reviewer #1: Yes

Reviewer #2: Yes

3. Has the statistical analysis been performed appropriately and rigorously? 

Reviewer #1: Yes

Reviewer #2: Yes

4. Have the authors made all data underlying the findings in their manuscript fully available?

Reviewer #1: (No Response)

Reviewer #2: Yes

5. Is the manuscript presented in an intelligible fashion and written in standard English?

Reviewer #1: (No Response)

Reviewer #2: Yes

6. Review Comments to the Author

Reviewer #1: I did not capture this in my initial review; however, it would be interesting to include a supplementary figure with panels showing the differences among neutrophil subsets within (A) MLE, (B) ME, (C) BL, and (D) HA. For example, in (A), Immature vs mature inactive vs mature active vs aged vs atypical aged.

Reviewer #2: (No Response)

7. PLOS authors have the option to publish the peer review history of their article (what does this mean?). If published, this will include your full peer review and any attached files.

Reviewer #1: No

Reviewer #2: No

---

## [Editor Report · Acceptance letter]

PONE-D-25-56348R1

PLOS One

Dear Dr. Forconi,

I'm pleased to inform you that your manuscript has been deemed suitable for publication in PLOS One. Congratulations! Your manuscript is now being handed over to our production team.

Kind regards,

on behalf of

Dr. Srinivasa Reddy Bonam

Academic Editor

PLOS One